# Comparison of Chemical and Sensory Profiles between Cabernet Sauvignon and Marselan Dry Red Wines in China

**DOI:** 10.3390/foods12051110

**Published:** 2023-03-05

**Authors:** Xixian Song, Weixi Yang, Xu Qian, Xinke Zhang, Mengqi Ling, Li Yang, Ying Shi, Changqing Duan, Yibin Lan

**Affiliations:** 1Center for Viticulture and Enology, College of Food Science and Nutritional Engineering, China Agricultural University, Beijing 100083, China; 2Key Laboratory of Viticulture and Enology, Ministry of Agriculture and Rural Affairs, Beijing 100083, China; 3School of Biology and Food Engineering, Changshu Institute of Technology, Changshu 215500, China; 4College of Food Science and Nutritional Engineering, Beijing University of Agriculture, Beijing 102206, China; 5“The Belt and Road” International Institute of Grape and Wine Industry Innovation, Beijing University of Agriculture, Beijing 102206, China

**Keywords:** Marselan, Cabernet Sauvignon, aroma compounds, polyphenols, sensory analysis

## Abstract

The differences in chemical and sensory characteristics between Marselan and Cabernet Sauvignon in China were investigated with gas chromatography–mass spectrometry (GC-MS) and high-performance liquid chromatography–triple quadrupole mass spectrometry (HPLC-QqQ-MS/MS), combined with color parameters and sensory data. The paired *t*-test results showed that terpenoids, higher alcohols, and aliphatic lactones were significantly different according to the grape variety. Meanwhile, terpenoids could be considered as marker aroma compounds to distinguish Marselan wines from Cabernet Sauvignon, which could explain the distinct floral note in Marselan wines. The mean concentrations of the mv-vsol, mv-vgol, mv-vcol, mvC-vgol, mv-v(e)cat, mvC-v(e)cat, mv-di(e)cat, and cafA were higher in Marselan wines than Cabernet Sauvignon wines, and these compounds might confer Marselan wines with a deeper color, more red hue, and higher tannin quality. The phenolic profiles of Marselan and Cabernet Sauvignon wines were influenced by the winemaking process, mitigating the varietal differences. As for sensory evaluation, the intensities of herbaceous, oak, and astringency of Cabernet Sauvignon were more pronounced than Marselan, whereas the Marselan wines were characterized by a high color intensity and more redness, together with floral, sweet, and roasted sweet potato attributes, and tannin roughness.

## 1. Introduction

Wine typicity is determined by its appearance, aroma, and taste, which are affected by grape variety, region, viticultural practice, and vinification technique [1]. Among them, grape variety can be considered one of the key factors which can directly influence the wine composition, such as sugars, acids, aroma compounds, phenolic compounds, polysaccharides, etc. For example, the varietal aroma of a specific grape variety could be ascribed to the special structured aromatic compounds which are already present in the grape, such as terpenoids responsible for the floral note of Muscat, thiols responsible for the passion fruit aroma of Sauvignon Blanc, and methoxypyrazines responsible for the green pepper note of Cabernet Sauvignon, Cabernet Franc, Cabernet Gernischt, and Carmenere. In addition, some grape varieties have higher concentrations of tannin, which will contribute to the strong astringency of wines such as Nebbiolo and Sangiovese. Therefore, consumers can recognize the variety of wine through sensory tasting, although the wines of the same grape variety from different regions are made in various vinification methods.

According to the report of the International Organization of Vine and Wine (OIV) (2020), in China, the planting area of wine grapes reached 783,000 hectares, which was dominated by red grape varieties, especially Cabernet Sauvignon, accounting for approximately 80% of reds [2]. Except for cool climate regions (e.g., Northeast China), Cabernet Sauvignon is widely planted in most regions for dry red wine production. In the last three decades, various varieties have been introduced into China to meet the demand for the production of premium wines of typical characteristics which can well express the various terroirs of different regions. Among them, Marselan (*Vitis vinifera* L.), a hybrid cultivar of Cabernet Sauvignon and Grenache [3], was introduced to China in 2001. Although the total planting area of Marselan grape is not large, this variety has become popular and the planting area has been increasing year by year due to its premium quality and good resistance to diseases [4]. At present, many growers and wineries choose to remove the vineyards of Cabernet Sauvignon and other varieties and plant these vineyards with Marselan grapes. Nevertheless, the specific performance of Marselan in China is unclear. Whether Marselan can become a new Chinese brand variety similar to Cabernet Sauvignon still needs to be researched and explored.

In the last decade, many researchers started focusing on the studies of the influence of viticultural practices on Marselan grape compositions and the impact of vinification parameters on the flavor characteristics of its wines. The key odorants of Marselan dry red wine were identified with gas chromatography–olfactometry (GC-O) in combination with aroma extract dilution analysis (AEDA) [5]. The authors found that, in Marselan wine, *β*-damascenone had the highest FD factors, followed by eugenol, 2,3-butanedione, citronellol, 4-hydroxy-2,5-dimethyl-3(2H)-furanone, phenethyl acetate, guaiacol, and 2-methoxy-4-vinylphenol and reported that Marselan wine was characterized by blackberry, green pepper, honey, raspberry, caramel, smoky, and cinnamon aromas [5]. In our previous study, the chemical and sensory characteristics of young Marselan red wines from five regions of China were investigated [6]. The Marselan wines in Jiaodong Peninsula and Bohai Bay regions of China had lower average concentrations of *β*-citronellol, geraniol, (*E*)-*β*-damascenone, isoamyl acetate, octanoic acid, decanoic acid, ethyl decanoate, etc.) than Xinjiang, Loess Plateau, and Huaizhuo Basin regions. Meanwhile, Marselan wines from Xinjiang were discriminated against due to their higher concentrations of several flavonols. However, the difference in flavor characteristics between Cabernet Sauvignon and Marselan wines in China is still unclear.

In this study, the differences in the aromatic and phenolic compounds and sensory profiles between Cabernet Sauvignon and Marselan wines in China were investigated based on the quantitative data obtained from HPLC-QqQ-MS/MS and GC-MS, in combination with color parameters and quantitative descriptive analysis (QDA).

## 2. Materials and Methods

### 2.1. Wine Samples

In this study, 12 Cabernet Sauvignon and 12 Marselan dry red wines were collected from 12 wineries in 5 wine-producing regions in China, including the Jiaodong Peninsula, the Huaizhuo Basin, the Loess Plateau, the Eastern Foothill of Helan Mountain, and Xinjiang. For each winery, both Cabernet Sauvignon and Marselan wines at the same vintage were made with the same vinification technique. Wine information was summarized in Table 1. The oenological parameters of Cabernet Sauvignon and Marselan dry red wines from five producing regions are shown in Appendix A.

### 2.2. Reagents

Analytical-grade chemicals, including sodium chloride, sodium hydroxide, tartaric acid, anhydrous sodium sulfate, and glucose, were purchased from Beijing Chemical Works (Beijing, China). Chromatographic-grade reagents, including methanol, ethanol, dichloromethane, formic acid, acetonitrile, and acetic acid, were purchased from Fisher (Fairlawn, NJ, USA) and Honeywell (Marris Township, NJ, USA). Malvidin-3-O-glucoside and other reference standards of non-anthocyanin phenolic compounds were purchased from Sigma-Aldrich (St. Louis, MO, USA), ChromaDex (Irvine, CA, USA), and Extrasynthese (Genay, France). Reference standards of aroma compounds and C6-C24 n-alkanes were obtained from Fluka (Buchs, Switzerland) and Sigma-Aldrich (St. Louis, MO, USA). Cleanert PEP-SPE resins (1000 mg/12 mL) were purchased from Bonna-Agela Technologies Inc. (Tianjin, China).

### 2.3. Quantitation of Aroma Compounds

#### 2.3.1. Headspace–Solid-Phase Microextraction–Gas Chromatography–Mass Spectrometry (HS-SPME-GC-MS)

The extraction and quantitation of aroma compounds were performed with HS-SPME-GC-MS according to our published method [7]. The analysis system was an Agilent 7890 GC gas chromatograph combined with an Agilent 5975 MS mass spectrometer. A 5 mL wine sample was mixed with 10 μL of 4-methyl-2-pentanol (internal standard, 1.0086 g/L) and 1 g of NaCl. The mixture was placed into a 20 mL vial capped with a PTFE-silicon septum. The volatile compounds of wine samples were extracted with automated headspace solid-phase microextraction using CTC CombiPAL (CTC Analytics, Zwingen, Switzerland). The sample vial was heated and balanced at 40 °C for 30 min, and the heating tank was rotated at 500 rpm. Then, the activated SPME fiber (50/30 μm DVB/CAR/PDMS, Supelco, Bellefonte, PA., USA) was inserted to extract the volatile compounds at 40 °C for 30 min. Then, the fiber was immediately inserted into the GC injector. The sample was injected at 250 °C in a split mode with a ratio of 5:1. The capillary column was HP-INNOWAX (60 m × 0.25 mm × 0.25 µm, J&W Scientific, Folsom, CA, USA). The carrier gas was high-purity helium, and the flow rate of carrier gas was 1 mL/min. The oven temperature was held at 50 °C for 1 min after injection, then programmed to 220 °C at a rate of 3 °C/min and held at 220 °C for 5 min. Qualitative and quantitative information for the major aroma compounds is listed in Appendix A.

#### 2.3.2. Solid-Phase Extraction–GC–Triple Quadrupole MS/MS (SPE-GC-QqQ-MS/MS)

Aliphatic lactones in wines were extracted with the SPE method according to our published method [8]. A Cleanert PEP-SPE (1000 mg/12 mL) column was first activated by 10 mL of dichloromethane, 10 mL of methanol, and 10 mL of 11% (*v*/*v*) ethanol. Then, 20 mL wine samples were added, followed by 10 mL of deionized water to remove sugar, acid, and macromolecule polar compounds; then, 15 mL dichloromethane, the eluent, was added with 1 g of anhydrous sodium sulfate to remove water as much as possible. Finally, the volatile extract was concentrated to 500 μL under a steam of nitrogen and filtered with a 0.45 μm filter membrane.

An Agilent Intuvo 9000 GC system equipped with an Agilent 7693 autosampler and an Agilent 7010B GC–MS/MS triple quadrupole (Agilent Technologies, Palo Alto, CA, USA) was used for quantitative analysis of aliphatic lactones [8]. A column HP-5MS UI 30 m × 0.25 mm × 0.25 μm (Agilent Ultra Inert GC column) was used to separate analytes. The extract (1 μL) was injected in splitless mode, and splitless time was 1 min. Helium was used as the carrier gas (1.0 mL/min) and the quenching gas (2.25 mL/min), and nitrogen was the collision gas (1.5 mL/min). The oven temperature program was as follows: 40 °C, held for 1 min, then increased to 160 °C at a rate of 5 °C/min, and finally increased to 220 °C at a rate of 30 °C/min and held for 1 min. After each analytical run, the post-run was performed at 280 °C for 2 min. A solvent delay of 5 min was performed for each analysis to prevent instrument damage. The temperatures of the injector, ion source, transfer line, and quadrupole 1 and 2 were 250 °C, 230 °C, 300 °C, 150 °C, and 150 °C, respectively. The mass spectrometer was operated in electron ionization (EI) mode at 70 eV with MRM. Data acquisition and analyses were performed using the MassHunter Workstation software version B.10.00, supplied by the manufacturer (Agilent Technologies, Palo Alto, CA, USA). Qualitative and quantitative information for the aliphatic lactones is listed in Appendix A.

### 2.4. Colorimetric Measurements

The chromatic parameters of wines were measured with a UV-visible spectrophotometer (Shimadzu UV-2450, Shimadzu Co., Kyoto, Japan), recording the wine absorbance spectra (380–700 nm) with a 1 nm wavelength of interval. Wine samples were filtered through cellulose filters (0.45 μm; Jinteng company, Tianjin, China) and placed in a 2 mm path length glass cuvette with distilled water as a reference. Each wine was analyzed in triplicate. The CIELAB parameters, expressed in terms of the rectangular (*L**, *a**, *b**) and cylindrical (*L**, *C*_ab_*, *h_ab_*) color coordinates, were calculated according to a previously published method [9].

### 2.5. Quantitation of Phenolic Compounds by HPLC-QqQ-MS/MS

Phenolic compounds were identified and quantified using HPLC-QqQ-MS/MS based on our previously established methods with slight modifications [10,11]. The HPLC was Agilent 1200 series, combined with a 6410B triple tandem quadrupole mass spectrometer. The column was a Poroshell 120 EC-C18 (150 × 2.1 mm, 2.7 μm). The samples were filtered through a 0.22 μm filter before analysis. The mobile phases used for elution were A (0.1% formic acid aqueous solution) and B (50/50 methanol acetonitrile solution with 0.1% formic acid). The elution procedure of phenolic compounds was 0–10 min, 10% B; 10–15 min, 100% B. The flow rate of the mobile phase was 0.4 mL/min, the column temperature was maintained at 55 °C, and the injection volume was 5 μL. The elution program of anthocyanin derivatives was 1 min, 100% A; 3 min, 25% B; 15 min, 30% B; and 20 min, 100% B. The posting time was for 5 min. The flow rate of the mobile phase was 0.3 mL/min, the column temperature was controlled at 55 °C, and the injection volume was 10 μL. An electrospray ionization source was used with a spray voltage of 4 kV in negative and positive modes for the non-anthocyanins and anthocyanins, respectively. The temperatures of the ion source and drying gas (N2) were 150 °C and 350 °C, respectively. The gas flow rate was 12 L/h and the nebulizer pressure was 35 psi. Multiple reaction monitoring (MRM) was selected as a scanning mode for both identification and quantification.

The qualitative of phenolic compounds was achieved based on their mass information and retention time. Monomeric anthocyanins and their derivatives were quantified based on the calibration curve of malvidin-3-*O*-glucoside and non-anthocyanins were quantified according to the calibration curves of their reference standards. Qualitative and quantitative information for the phenolic compounds were listed in Appendix A.

### 2.6. Sensory Analysis

Sensory analysis was carried out in a professional tasting laboratory, and the ambient temperature was 20 °C. There are 20 individual booths in the tasting laboratory to separate the panelists. The wine samples were presented in the wine-tasting glass (ISO 3591:1977) of the International Organization for Standardization (ISO). In all cases, approximately 30 mL wine samples were served in the glass labeled with three-digit random numbers, and the glasses were served in a randomized order. Panelists were not informed about the information on the wine samples.

A total of 24 wines including 12 Cabernet Sauvignon and 12 Marselan dry red wines were evaluated by 20 trained panelists (7 males and 13 females; 21–28 years old) who were students or staff members or faculties from the Center for Viticulture and Enology. All panelists participated in four training sessions (2 h each). In the first session, all panelists were asked to sniff all wine samples used in this study and generated a list of descriptors to describe the aroma perceptions of these wines. In the second and third sessions, all panelists discussed the list of descriptors and generated the final lexicon terms, including eight aroma attributes (including red fruit, black fruit, fresh fruit, floral, herbaceous, sweet, and oak) and two oral attributes (including astringency and tannin quality). Reference standards of each aroma attribute were prepared from Le Nez du Vin (Jean Lenoir, Provence, France), or natural products (fruits, juices, vegetables, etc.) (Appendix A) and were used to train the panelists. In the fourth session, panelists were asked to rate the intensities of eight reference attributes and representative wines.

During the formal sessions, quantitative descriptive analysis (QDA) was carried out to evaluate the wine samples according to a previous study [7]. All panelists were asked to rate the intensity of 8 aroma attributes and score the 2 oral attributes on an 11-point scale for each wine (0 = very low intensity, 10 = strong intensity). The total wine samples were divided into four sub-sessions for scoring, which lasted for a total of 2.5 h, with a 5 min break in each sub-session. We used PanelCheck v1.4.2 software (https://www.panelcheck.com, accessed on 1 September 2022) to make sure there was no significant difference among panelists (*p* > 0.05) in every formal session. The average score of each sensory attribute was calculated using a boxplot analysis after outlier panelist inspection for further statistical analysis [12].

### 2.7. Statistical Analysis

We used SPSS (IBM SPSS Statistics 24) to make sure that all data coincided with normal distribution in the Shapiro–Wilk test (*p* > 0.05). The paired *t*-test analysis was used to compare the oenological parameters aroma compounds, phenolic compounds and sensory data between Cabernet Sauvignon and Marselan from five different regions. The partial least squares–discriminant analysis (PLS-DA) and principal component analysis (PCA) were used to investigate the difference in aroma compounds and phenolic compounds between Cabernet Sauvignon and Marselan. The PCA of aroma compounds and the PLS-DA of phenolic compounds were shown in Appendix A. The paired *t*-test analysis, PLS-DA, and PCA were performed using XLSTAT 2019 statistical software (Addinsoft, New York, NY, USA).

## 3. Results and Discussion

### 3.1. Aroma Compounds

A total of 64 aroma compounds were identified and quantified in Cabernet Sauvignon and Marselan dry red wine (Table 2). The aroma activity value (OAV) of each compound is calculated according to the threshold value, the ratio of the aroma compound threshold value, and the concentration of the compound. When the OAV of the aroma compound in wine is greater than 1, the compound is considered to have an important contribution to the overall aroma of wine [13,14]. As shown in Table 2, a total of 18 aroma compounds had OAVs greater than 1 identified in both 2 varieties of wines, including ethyl acetate, ethyl butanoate, ethyl hexanoate, ethyl octanoate, ethyl decanoate, ethyl 2-methylbutanoate, ethyl 3-methylbutanoate, ethyl cinnamate, isoamyl acetate, isoamyl alcohol, 2-phenylethyl alcohol, methionol, isovaleric acid, octanoic acid, *β*-damascenone, *β*-ionone, and (*Z*)-oaklactone. *γ*-Butyrolactone displayed OAVs greater than one only in Cabernet Sauvignon wines.

The paired *t*-test was carried out to identify the differences in aroma compounds between the two varieties of wines. It was shown that twelve volatile compounds, mainly terpenoids, higher alcohols, and aliphatic lactones, were significantly different according to the grape variety. Among the aroma compounds, three terpenoids (linalool, terpinen-4-ol, and α-terpineol), three lactones (*γ*-butyrolactone, *δ*-octanolactone, and *γ*-undecalactone), three higher alcohols (isoamyl alcohol, 2-phenylethyl alcohol, and methionol), isoamyl hexanoate, ethyl hexadecanoate, and isovaleric acid showed significant differences in concentration (Table 2).

It is well known that grape-derived aroma compounds contribute to the typicity aromas of a specific grape variety and could be considered the marker chemicals responsible for the differentiation from other varieties. Among the different aroma compounds between the two varieties, terpenoids, and aliphatic lactones were two important categories of grape-derived aroma compounds. In this study, the total concentration of terpenoids was approximately 1.5-fold higher in the Marselan wines than in Cabernet Sauvignon, with a value of 21.79 μg/L, which was mainly dominated by linalool, terpinen-4-ol, and α-terpineol (Table 2). The results were consistent with our previous study, which found higher concentrations of monoterpenes in young Marselan wine compared with neutral grape varieties [6]. To our knowledge, these compounds are considered varietal aroma compounds, which are important contributors to the aroma of wines made from Muscat varieties (e.g., Muscat of Alexandria and Muscat Blanc à Petit Grain) and correlate with the floral aroma in wines [15]. The differences between the wines of the two varieties were anticipated due to the genetic differences affecting the final content of the aroma compounds in wine. However, both Cabernet Sauvignon and Grenache, the parents of the Marselan grape, are considered neutral varieties, characterized by their relatively low content of terpenes [16]. Interestingly, previous research reported that Grenache wine showed maximum flavor dilution values in the case of terpenes in a comparative GC-olfactometry (GC-O) study between young wines made from Merlot, Cabernet Sauvignon, and Grenache grape varieties [17]. Therefore, it seemed that the levels of terpenes in Marselan wines could be associated with the character inheritance from the Grenache grape, which required confirmation by further studies. From the perspective of aroma contribution, the relatively high concentration of terpenes could be related to the intense floral note in young Marselan wine. For instance, the average concentration of linalool (mean average con. 7.82 μg/L) in Marselan wines could reach half of its odor threshold (15 μg/L in water/ethanol solution) [18] and might contribute to wine with flowery and muscat notes.

Previous studies reported that aliphatic lactones are associated with a variety of aroma characteristics in wine, such as peach, apricot, coconut, and dried fruits [8]. In this study, concentrations of *δ*-octalactone, *γ*-undecalactone, and *γ*-butyrolactone were higher in the Cabernet Sauvignon wines than Marselan wines. This result may be ascribed to complex factors, such as the harvesting date, the thickness of the skin, and the degree of dehydration [19], which could not be concluded in this study. For example, a previous study showed that red Merlot and Cabernet Sauvignon wines made from berries infected and withered by *Plasmopara viticola* contained large amounts of *γ*-lactones [20].

For fermentation-derived aroma compounds, three higher alcohols (isoamyl alcohol, 2-phenylethyl alcohol, and methionol), isoamyl hexanoate, ethyl hexadecanoate, and isovaleric acid had significantly higher mean concentrations in Cabernet Sauvignon wines than those of Marselan wines. However, the mean concentrations of isoamyl hexanoate, ethyl hexadecanoate, were lower than those of Marselan dry red wine. These compounds mainly originated from yeast and bacterial metabolism on grape compositions, such as sugar, amino acids, and fatty acids [21]. The variation in these grape compositions could contribute to the differences in fermentation-derived aroma compounds between the two varieties. However, further comprehensive studies are needed to investigate the differences in these chemical components.

PCA exploratory data analysis was carried out and found that the volatile compounds were more influenced by the variety (Appendix A). Then, an overview of the differences in aroma compounds between Cabernet Sauvignon and Marselan was obtained by partial least squares–discriminant analysis (PLS-DA). The optimal number of factors for the calibration was selected based on the PRESS. As a result, a total of 16 factors of the PLS model were selected. As shown in Figure 1, a clear separation between the two varieties was obtained by a reliable PLS-DA model. As shown in Figure 1, the Cabernet Sauvignon wines were mainly located on the positive axis of the t1. The Marselan wines were mainly located on the negative axis of the t1. The distribution of the target metabolites can reflect the contribution between the two varieties of wines. The first and second components (t1 and t2), accounting for 32.7% of the total variance (total amount of variance explained in the x matrix [R^2^X] [cumulative] = 0.326, the total amount of variance explained in the y matrix [R^2^Y] = 0.932, and Q^2^ [cumulative] = 0.551). Isoamyl alcohol, 2-phenylethyl alcohol, isovaleric acid, methionol, *γ*-butyrolactone, and 1-nonanol revealed a high positive correlation with the first component (t1), whereas *δ*-octalactone, *α*-terpineol, methyl salicylate, linalool, ethyl hexadecanoate, *γ*-decalactone, *δ*-decalactone, and 4-terpineol revealed a strong negative correlation with t1. All Marselan wines are located on the negative axis of t1, and all Cabernet Sauvignon wines are located on the opposite axis of t1. Based on the variable influence on projection (VIP) score above 1, the predominant aroma compounds responsible for the discriminations were isoamyl alcohol, followed by methionol, isovaleric acid, *δ*-octalactone, 2-phenylethyl alcohol, *α*-terpineol, *γ*-butyrolactone, isoamyl hexanoate, ethyl hexadecanoate, linalool, *γ*-decalactone, 1-nonanol, 4-terpineol, ethyl cinnamate, phenol, methyl salicylate, 1-hexanol, ethyl nonanoate, and ethyl heptanoate. Among these compounds, the varietal aroma compounds, especially terpenoids, could be considered marker aroma compounds to distinguish Marselan wines from Cabernet Sauvignon, which could explain the distinct floral note in young Marselan wines.

**Table 2 foods-12-01110-t002:** The paired *t*-test results of aroma compounds (μg/L) in Cabernet Sauvignon and Marselan dry red wines from five regions in China.

Compound	Class ^a^	Threshold (μg/L) ^b^	Cabernet Sauvignon				Marselan				*p*-Value
			Minimum	Maximum	Mean	OAV ^c^	Minimum	Maximum	Mean	OAV	
1-Hexanol	C6	8000 [1]	1810	4370	2792	0.35	1371	3326	2166	0.27	0.075
(*Z*)-3-Hexenol	C6	400 [1]	131.2	289	182.9	0.46	112.5	221.6	179.9	0.45	0.822
Linalool	T	15 [1]	3.84	7.75	5.56	0.37	3.89	10.63	7.82	0.52	**0.014 ***
Terpinen-4-ol	T	5000 [2]	3.45	6.05	3.89	<0.01	3.45	8.5	5.22	0	**0.014 ***
α-Terpineol	T	250 [3]	1.36	6.77	4.69	0.02	1.36	12.63	8.75	0.04	**0.032 ***
*β*-Damascenone	N	0.05 [1]	2.35	8.93	5.16	103.2	2.7	9.78	5.09	101.8	0.873
*β*-Ionone	N	0.09 [3]	0.62	3.2	0.94	10.44	0.62	0.62	0.62	6.89	0.351
Isobutanol	H	40,000 [1]	20,875	33,091	27,821	0.70	20,422	40,174	26,741	0.67	0.685
Isoamyl alcohol	H	30,000 [1]	443,808	599,200	532,549	17.75	342,445	478,139	408,934	13.63	**0.004 ****
1-Heptanol	H	2500 [2]	18.71	58.77	32.78	0.01	14.16	58.49	28.75	0.01	0.52
2-Ethylhexanol	H	-	2.81	10.59	5.49	-	2.55	25.18	8.27	-	0.383
1-Octanol	H	800 [4]	13.58	24.76	17.21	0.02	11.16	24.53	17	0.02	0.903
2,3-Butanediol	H	150,000 [4]	73,548	120,706	91,623	0.61	70,200	105,393	85,503	0.57	0.384
1-Nonanol	H	600 [5]	8.11	13.89	10.05	0.02	4.14	10.82	7.85	0.01	0.107
1-Decanol	H	400 [3]	3.58	5.77	4.55	0.01	3.36	5.69	4.47	0.01	0.793
Benzyl alcohol	H	900,000 [2]	436	727.8	562.2	<0.01	451.9	1271	633.5	0	0.466
2-Phenylethyl alcohol	H	10,000 [1]	30,013	58,484	46145	4.61	13,145	44,894	30,740	3.07	**0.027 ***
Ethyl butanoate	EEFA	20 [1]	323.9	732.6	505.8	25.29	337.4	739.8	509.1	25.46	0.887
Ethyl hexanoate	EEFA	5 [1]	637.3	1864	1306	261.2	552.4	2383	1286	257.2	0.888
Ethyl octanoate	EEFA	580 [2]	939.1	2506	1637	2.82	859.6	3210	1608	2.77	0.887
Ethyl decanoate	EEFA	200 [3]	105.3	582	295.3	1.48	75.18	784.7	319.3	1.6	0.685
Ethyl dodecanoate	EEFA	500 [2]	0.3	1.18	0.63	<0.01	0.29	1.92	0.9	0	0.129
Ethyl hexadecanoate	EEFA	1000 [2]	19.75	68.69	36.31	0.04	34.09	106.9	64.3	0.06	**<0.001 *****
Ethyl 2-methylbutanoate	EEBA	1 [1]	0.46	100.2	54	54.0	0.46	95.36	41.63	41.63	0.116
Ethyl 3-methylbutanoate	EEBA	3 [1]	2.86	152.2	77.83	25.94	2.86	149.1	68.95	22.98	0.292
Ethyl acetate	EEOA	75,000 [1]	213,390	4,088,045	289,335	3.86	199,877	506,477	323,956	4.32	0.1
Ethyl heptanoate	EEOA	220 [2]	4.75	7.76	5.69	0.03	4.44	6.18	5.13	0.02	0.199
Ethyl lactate	EEOA	150,000 [2]	71914	195,240	109,931	0.73	67,826	147,,835	107,157	0.71	0.8
Ethyl nonanoate	EEOA	-	1.99	3.33	2.46	-	1.6	2.68	2.14	-	0.073
Ethyl 2-hydroxy-4-methylpentanoate	EEOA	300 [6]	128.7	255.8	183.5	0.61	104.9	401.8	186	0.62	0.925
Ethyl 2-furoate	EEOA	16,000 [3]	13.66	25.94	19.57	<0.01	11	29.08	18.62	0	0.372
Dithyl succinate	EEOA	200,000 [7]	2348	28,121	18,801	0.09	3700	32,033	19,184	0.1	0.832
Ethyl benzeneacetate	EEOA	-	4.18	9.31	6.58	-	2.5	9.88	5.84	-	0.079
Ethyl cinnamate	EEOA	1 [1]	51.36	411.6	114.4	114.4	33.33	46.29	36.88	36.88	0.107
Methyl octanoate	OE	200 [8]	3.91	6.58	4.86	0.02	3.69	6.31	4.7	0.02	0.551
Methyl salicylate	OE	-	6.99	11.27	8.42	-	7.97	19.23	10.62	-	0.163
Isoamyl hexanoate	OE	-	3.63	4.75	4.15	-	3.11	4.36	3.67	-	**<0.001 *****
Isoamyl octanoate	OE	125 [3]	7.55	16.57	10.95	0.09	6.66	15.87	10.07	0.08	0.255
Isoamyl acetate	HA	30 [1]	680	1948	1135	37.83	484.8	3074	951.4	31.71	0.605
Hexyl acetate	HA	1000 [9]	11.22	30.47	17.66	0.02	10.19	62.49	19.28	0.02	0.782
*β*-Phenethyl acetate	HA	250 [1]	2.86	4.04	3.22	0.01	2.5	6.42	3.27	0.01	0.887
Furfural	F	14,100 [3]	316.5	4557	1398	0.10	191.4	5090	1250	0.09	0.858
5-Methylfurfural	F	20,000 [7]	106.7	620.4	314.5	0.02	106.7	667.6	233.5	0.01	0.401
Acetic acid	A	200,000 [1]	0.11	0.19	0.14	<0.01	0.08	0.2	0.13	0	0.927
Isobutyric acid	A	2300 [3]	1072	2998	1844	0.80	1042	2745	1467	0.64	0.245
Isovaleric acid	A	33.4 [3]	1092	1729	1288	38.56	852	1097	978.3	29.29	**0.007 ****
Octanoic acid	A	500 [3]	2160	2753	2307	4.61	1902	4021	2440	4.88	0.55
*n*-Decanoic acid	A	1000 [3]	236.3	279.2	252.5	0.25	211.49	482.6	266.6	0.27	0.656
*γ*-Butyrolactone	L	20,000 [4]	17,079	27,552	22,754	1.14	10,273	23,094	17,719	0.89	**0.011 ***
Pantolactone	L	2000 [4]	304	634.4	480.7	0.24	314.2	815.1	537.9	0.27	0.429
*γ*-Hexalactone	L	13,000 [3]	16.04	245.5	67	0.01	0.4	0.4	0.4	0	0.403
*γ*-Octalactone	L	7 [10]	0.72	78.13	10.55	1.51	0.82	3.99	1.44	0.21	0.36
*δ*-Octalactone	L	400 [11]	2.57	7.34	6.09	0.02	6.71	10.17	8.22	0.02	**0.003 ****
*γ*-Nonalactone	L	30 [3]	4.2	11.74	8.53	0.28	3.81	10.95	8.04	0.27	0.41
*γ*-Decalatone	L	0.7 [10]	0.33	0.71	0.45	0.64	0.48	0.69	0.56	0.8	0.067
*δ*-Decalactone	L	100 [10]	3.59	6.29	4.9	0.05	3.99	11.5	5.98	0.06	0.145
*γ*-Undecalactone	L	60 [7]	0.27	0.35	0.3	0.01	0.28	0.35	0.31	0.01	**0.025 ***
*γ*-Dodecalactone	L	7 [10]	0.55	0.68	0.6	0.09	0.57	0.71	0.62	0.09	0.056
*δ*-Dodecalactone	L	-	2.96	4.9	3.75	-	2.66	6.58	3.96	-	0.466
Sotolon	L	5 [1]	0.66	3.43	1.97	0.39	0.62	5.39	2.53	0.51	0.205
Benzaldehyde	O	2000 [8]	11.93	41.34	23.44	0.01	10.84	205.2	48.91	0.02	0.281
Methionol	O	1000 [3]	1976	4036	2824	2.82	645.6	2610	1691	1.69	**0.009 ****
(*Z*)-Oaklactone	O	67 [2]	172.2	558.2	424	6.33	122.9	799.5	439.9	6.57	0.824
Phenol	O	-	14.93	53.6	26.71	-	<0.01	27.88	16.49	-	0.183

^a^ C6, C6 alcohols; T, terpenoilds; N, C13-norisoprenoids; H, higher alcohols; HA, higher alcohol acetates; EEFA, ethyl esters of straight-chain fatty acids; EEBA, ethyl ester of branched acids; EEOA, ethyl esters of other acids; et al., OE, other esters; F, furfural; A, acids; L, lactones; O, other compounds. ^b^ [1] Guth, 1997; [2] Zea, Moyano, Moreno, Cortes, and Medina, 2001; [3] Ferreira, López, and Cacho, 2000; [4] Pozzatti et al., 2006; [5] Tao and Zhang, 2010; [6] Falcao, Lytra, Darriet, and Barbe et al., 2012; [7] Culleré, Escudero, Cacho, and Ferreira, 2004; [8] Jiang and Zhang, 2010; [9] Chaves, Zea, Moyano, and Medina, 2007; [10] Ferreira, Jarauta, Ortega, and Cacho, 2004; [11] Loscos, Hernandez-Orte, Cacho, and Ferreira, 2007; ‘-’ represents no reported odor threshold. ^c^ OAV, odor activity value, which is calculated by dividing the mean concentration of aroma compounds by odor thresholds. *, **, *** Significant at *p* ≤ 0.05, 0.01 and 0.001, respectively.

### 3.2. Phenolic Compounds

A total of 72 phenolic compounds were identified and quantified in Cabernet Sauvignon and Marselan dry red wine. These phenolic compounds were classified into 10 categories, including monomeric anthocyanins, anthocyanin derivatives (vitisins, pinotins, acetaldehyde-bridged anthocyanin-flavan-3-ol condensation products (A-e-F), flavanyl-pyranoanthocyanins (A-v-F), and direct anthocyanin-flavan-3-ol condensation products (F-A/A-F)), flavan-3-ols, hydroxybenzoic acids, hydroxycinnamates, and flavonols (Table 3). A paired *t*-test was carried out to identify the differences in phenolic compounds between Cabernet Sauvignon and Marselan wines (Table 3). A total of 13 compounds showed significant differences, mainly including 9 anthocyanin derivatives and 3 phenolic acids.

Among the nine anthocyanin derivatives and three phenolic acids, the mean concentrations of the malvidin-3-*O*-glucoside-4-vinylsyringol (mv-vsol), malvidin-3-*O*-glucoside-4-vinylguaiacol (mv-vgol), malvidin-3-*O*-glucoside-4-vinylcatechol (mv-vcol), malvidin-3-*O*-coumaroylglucoside-4-vinylguaiacol (mvC-vgol), malvidin-3-*O*-glucoside-4-vinyl(epi)catechin (mv-v(e)cat), malvidin-3-*O*-coumaroylglucoside-4-vinyl(epi)catechin (mvC-v(e)cat), malvidin-3-*O*-glucoside-di(epi)catechin (mv-di(e)cat), and caffeic acid (cafA) were higher in Marselan wines than Cabernet Sauvignon wines (Table 3). The higher concentrations of these compounds may be the reason for the deeper color and red hue of Marselan. A relevant study observed more orange hue in anthocyanin with more methoxy substitution (malvidin, petunidin, and peonidin) than hydroxyl substitution (delphinidin and cyanidin) [22]. Moreover, the previous study also showed that aglycones (delphinidin, cyanidin, petunidin, peonidin, malvidin, and pelargonidin) differ from each other in the number of hydroxyl and methoxyl groups on the B-ring, such as the increasing hydroxylation conferring an increase in the blue tint and increasing methylation conferring an increase in the red tint [23,24]. In our study, the myricetin-3-*O*-galactoside (myr-3-gal) and quercetin-3-*O*-galactoside (que-3-gal) concentrations were relatively higher in Cabernet Sauvignon wines.

PCA exploratory data analysis was also carried out on the phenolic compounds and found that the phenolic compounds were less influenced by the variety. Then, an overview of the differences in phenolic compounds between Cabernet Sauvignon and Marselan was obtained by PLS-DA (Appendix A). However, the model was not reliable after parameter verification (Q^2^ = −0.291, R^2^X = 0.499, R^2^Y = 0.553). The principal component analysis of the phenolic compounds of the Marselan and Cabernet Sauvignon wines is shown in Figure 2. The PCA results explained 76.12% of the total variance. Meanwhile, the first dimension explained 50.05% of the total variance and the second dimension explained 16.27% of the total variance (Figure 2). As shown in Figure 2, the Marselan and Cabernet Sauvignon wines cannot be distinguished based on the phenolic compounds, which was different from the results based on aromatic compounds. However, interestingly, Marselan and Cabernet Sauvignon wines from the same winery had a similar distribution pattern (Figure 2). The reason for this phenomenon may be due to the nuances of the winery’s winemaking practice. For example, the concentration of free and acylated anthocyanins in the Merlot wine samples varies depending on the maceration technique [25]. Different maceration techniques could impact the phenolics profiles in the final wines. For example, previous researchers found that the total tannins and total anthocyanins in dry red wines increased with the increasing cold maceration time compared to the traditional process [26,27]. In this study, some varietal characteristics could be covered up by the in-house style of winemaking practice in each winery such as the extended maceration. Although Daudt et al. [28] found that the localization of the vineyards seemed to have more influences on the wine characteristics than the maceration type, the extended maceration increased the extraction of tannins resulting in greater color intensity and a greater quantity of anthocyanins. Moreover, Kennedy [29] found that higher temperatures increased the effectiveness of extraction and thus could reduce the time for maximum extraction. Nevertheless, previous studies showed that the chemical structure of phenolic compounds could significantly influence extractability due to the interactions with other matrix components [30]. For example, Boulton [31] researched that the formation of co-pigments would enhance the apparent solubility of both co-factors and anthocyanins and possibly reduce the extent of adsorption from the solution. Although, Shi et al. [32] reported that Marselan grape skins had significantly higher contents of anthocyanins than those of Cabernet Sauvignon grapes, and the Marselan grape skins were characterized by higher contents of two anthocyanins (malvidin and petunidin) and one non-anthocyanin (procyanidin trimer). However, in this study, the individual winemaking process of each winery (such as maceration technique, temperature, cap management, etc.) might pose an impact and influence the phenolic profile of wines of these two similar varieties, resulting in mitigating the differences between them.

### 3.3. Color Parameters Analysis and Sensory Analysis

The results of the paired *t*-test for 3 color parameters and 11 sensory attributes of all Cabernet Sauvignon and Marselan wines are shown in Table 4. Among the 13 parameters, *L**, *a**, herbaceous, and oak attributes showed significant differences between 2 varieties. The result showed that the *L** value of Cabernet Sauvignon was higher than that of Marselan, indicating that the color intensity of Marselan wine was pronounced than that of Cabernet Sauvignon. This phenomenon probably could be explained by the higher concentration of anthocyanin derivatives (including vitisins, pinotins, A-e-F, A-v-F and F-A/A-F) of Marselan wines in comparison with Cabernet Sauvignon wine (Table 3). These anthocyanin derivatives were in a low concentration though, they had almost one order of magnitude more molar fraction of pigmented molecules than their precusor thus making a sizeable contribution to wine color [33]. The *a** value of Marselan wines was higher than that of Cabernet Sauvignon wines, indicating that the Marselan wines had more red hue than Cabernet Sauvignon wines. Cabernet Sauvignon wines displayed stronger intensities in herbaceous and oak aromas. It is well known that methoxypyrazines are mainly responsible for the green pepper or herbaceous aromas in Cabernet Sauvignon wines [34]. In a recent study, although it was reported that Cabernet Sauvignon and Marselan grapes contained high levels of methoxypyrazines, the differences in methoxypyrazines concentrations between two varieties were affected by factors, such as vintage, and no conclusion was reached [35]. The intensities of astringency were higher in Cabernet Sauvignon wines, and the tannin quality was higher in Marselan wines. The previous study had shown that phenolic acids can made a difference in the quality and intensity of astringency [36,37]. Further studies are needed to confirm the contributions of methoxypyrazines to Marselan wines and the differences among various varieties. The difference in oak aroma may be caused by the application of different aging techniques, such as the origin and toasting degree of oak barrels [38].

In this study, although the *b**, floral, fresh fruit, dried fruit, and sweet attributes had no significant difference between two varieties, their *p*-values were relatively low, ranging from 0.075 to 0.175. Among them, all Marselan wines have floral and sweet that were higher in intensities than Cabernet Sauvignon wines, which could be associated with higher terpenes in Marselan wines mentioned above. There was no statistical difference in the performance of red fruit, black fruit, astringency, and tannin quality between two varieties of red wine.

## 4. Conclusions

The purpose of this study was to identify the key flavor compounds in Marselan dry red wine and compare them with Cabernet Sauvignon dry red wine, using HPLC-QqQ-MS/MS, GC-MS, combined color parameters, and QDA methods to analyze samples from five different wine regions in China. In conclusion, the paired *t*-test was carried out to identify the volatile compounds that terpenoids, higher alcohols, and aliphatic lactones were significantly different according to the grape variety. Among these compounds, terpenoids could be considered marker aroma compounds to distinguish Marselan wines from Cabernet Sauvignon, which could explain the distinct floral note in young Marselan wines. The mean concentrations of the mv-vsol, mv-vgol, mv-vcol, mvC-vgol, mv-v(e)cat, mvC-v(e)cat, mv-di(e)cat, and cafA were higher in Marselan wines than Cabernet Sauvignon wines, and these compounds might confer Marselan wines with a deeper color, more red hue, and higher tannin quality. Marselan and Cabernet Sauvignon wines from the same winery shared a similar polyphenol profile because the individual winemaking practice could have a great impact on the phenolic profile of wines of two varieties and mitigate the differences between them. Aroma compounds were more reflective of varietal differences, and polyphenols were mainly affected by the winemaking practices. The compounds of the two varieties were different, which was also reflected in the sensory characteristics. Meanwhile, the intensities of herbaceous and oak of Cabernet Sauvignon were higher than Marselan, and Cabernet Sauvignon wines had a greater intensity of astringency in China’s various producing areas. Moreover, Marselan wines’ dark color and red color were higher than Cabernet Sauvignon. The results obtained in this research could provide information to study how to modulate and express the sensory characteristics of Chinese Marselan dry red wine through the application of various winemaking practices.

## Figures and Tables

**Figure 1 foods-12-01110-f001:**
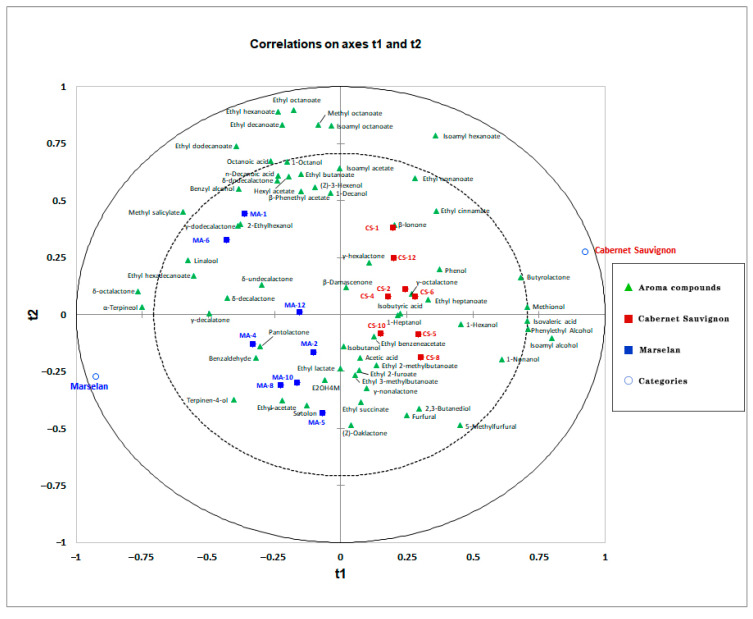
Partial least squares–discriminant analysis of Cabernet Sauvignon (CS) and Marselan (MA) dry red wines from five regions in China based on aroma compounds. Tringles represent the explanatory variables (X, aroma compounds). Circles represent the two studied categories (Cabernet Sauvignon and Marselan). Squares represent wine samples and sample names are in accordance with Table 1.

**Figure 2 foods-12-01110-f002:**
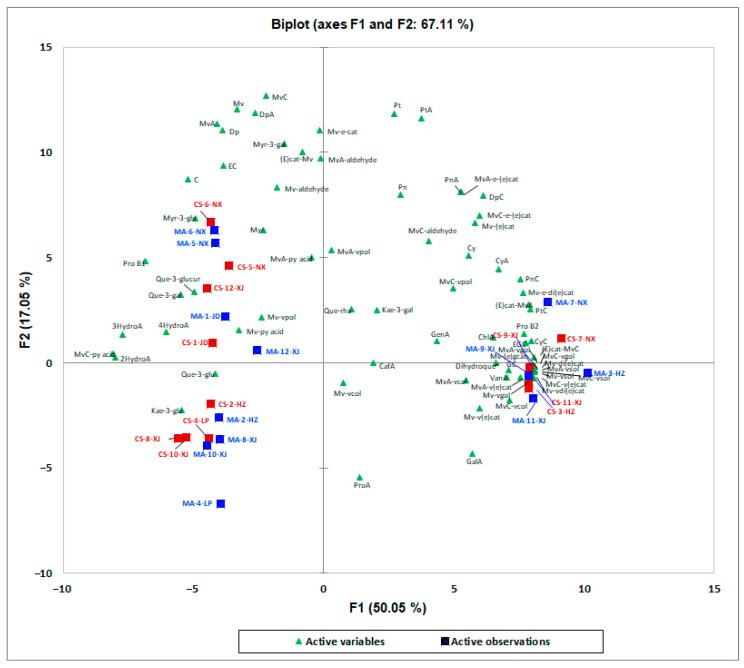
Principal component analysis of Cabernet Sauvignon and Marselan dry red wines from five regions in China based on phenolic compounds. CS, Cabernet Sauvignon; MA, Marselan; HT, Huangtu; LP, Loess Plateau; JD, Jiaodong; NX, Ningxia; XJ, Xinjiang.

**Table 1 foods-12-01110-t001:** Information of Cabernet Sauvignon and Marselan dry red wines from five regions in China.

Groups ^a^	Variety ^b^		Winery	Region	Vintage
1	CS-1	MA-1	Chateau Yunmo Greatwall	Jiaodong Peninsula	2016
2	CS-2	MA-2	Amethyst Manor	Huaizhuo Basin	2015
3	CS-3	MA-3	Martin Vineyard	Huaizhuo Basin	2017
4	CS-4	MA-4	Grace Vineyard	Loess Pplateau	2013
5	CS-5	MA-5	Huangkou Winery	Eastern Foothill of Helan Mountain	2017
6	CS-6	MA-6	Chateau Huahao	Eastern Foothill of Helan Mountain	2017
7	CS-7	MA-7	Chateau Huahao	Eastern Foothill of Helan Mountain	2018
8	CS-8	MA-8	Chateau Zhongfei	Xinjiang	2015
9	CS-9	MA-9	Chateau Zhongfei	Xinjiang	2016
10	CS-10	MA-10	Tiansai Vineyards	Xinjiang	2015
11	CS-11	MA-11	Tiansai Vineyards	Xinjiang	2016
12	CS-12	MA-12	CITIC Guoan Winery	Xinjiang	2017

^a^ Wines are divided into 12 groups according to their origin. ^b^ CS, Cabernet Sauvignon; MA, Marselan.

**Table 3 foods-12-01110-t003:** The paired *t*-test results of phenolic compounds (mg/L) in Cabernet Sauvignon and Marselan dry red wines from five regions in China ^a^.

No.	Compounds	Abbreviation	Cabernet Sauvignon			Marselan			*p*-Values
			Minimum	Maximum	Mean	Minimum	Maximum	Mean	
1	Cyanidin-glucoside	Cy	0.19	1.7	1	0.15	2.45	0.95	0.782
2	Cyanidin-3-*O*-acetylglucoside	CyA	0.17	1.87	0.94	0.12	2.41	0.9	0.791
3	Cyanidin-3-*O*-coumaroylglucoside (*cis* + *trans*)	CyC	0.12	1.55	0.65	0.11	1.56	0.67	0.653
4	Delphinidin-3-*O*-glucoside	Dp	3.47	22.77	9.97	0.98	21.23	9.3	0.577
5	Delphinidin-3-*O*-acetylglucoside	DpA	1.23	7.72	3.3	0.63	5.48	2.75	0.181
6	Delphinidin-3-*O*-coumaroylglucoside (*cis* + *trans*)	DpC	0.62	1.76	1.18	0.59	1.92	1.23	0.227
7	Malvidin-3-*O*-glucoside	Mv	23.98	172.81	79.26	1.82	206.07	81.88	0.692
8	Malvidin-acetylglucoside	MvA	10.78	99.13	37.19	1.61	86.46	32.06	**0.034 ***
9	Malvidin-coumaroylglucoside (*cis* + *trans*)	MvC	2.15	20.93	7.64	0.43	20.39	8.06	0.376
10	Peonidin-glucoside	Pn	0.56	4.89	2.37	0.28	4.68	2.08	0.348
11	Peonidin-acetylglucoside	PnA	0.42	3.23	1.58	0.12	3.35	1.34	0.153
12	Peonidin-coumaroylglucoside (*cis* + *trans*)	PnC	0.15	2.4	1.02	0.1	2.68	1.01	0.911
13	Petunidin-glucoside	Pt	0.74	8.87	3.8	0.23	13.04	4.19	0.382
14	Petunidin-acetylglucoside	PtA	0.4	4.12	1.92	0.11	5.12	1.83	0.624
15	Petunidin-coumaroylglucoside (*cis* + *trans*)	PtC	0.11	2	0.77	0.08	2.32	0.83	0.073
	**Total monomeric anthocyanins**	T-Anthocyanins	45.11	355.74	152.58	7.36	379.16	149.08	
16	Malvidin-3-*O*-glucoside-pyruvic acid	Mv-py acid	2.23	7.43	3.82	2.71	9.86	4.73	0.197
17	Malvidin-3-*O*-acetylglucoside-pyruvic acid	MvA-py acid	1.08	4.41	2.35	1.08	6.1	2.38	0.955
18	Malvidin-3-*O*-coumaroylglucoside-pyruvic acid	MvC-py acid	n.d.	0.09	0.06	n.d.	0.09	0.06	0.809
19	Malvidin-3-*O*-glucoside-acetaldehyde	Mv-aldehyde	0.99	5.76	2.71	0.68	20.55	3.76	0.460
20	Malvidin-3-*O*-acetylglucoside-acetaldehyde	MvA-aldehyde	0.56	4.01	1.87	0.28	5.68	1.63	0.535
21	Malvidin-3-*O*-coumaroylglucoside-acetaldehyde	MvC-aldehyde	0.29	1.97	1.07	0.17	3.71	1.13	0.840
	**Total vitisins**	T-Vitisins	5.16	23.68	11.89	4.92	46	13.7	
22	Malvidin-3-*O*-glucoside-4-vinylphenol	Mv-vpol	0.91	8.74	3.31	1.34	7.91	3.89	0.347
23	Malvidin-3-*O*-glucoside-4-vinylsyringol	Mv-vsol	0.11	1.53	0.59	0.11	1.54	0.61	**0.036 ***
24	Malvidin-3-*O*-glucoside-4-vinylguaiacol	Mv-vgol	0.29	1.68	0.85	0.33	1.67	0.96	**0.036 ***
25	Malvidin-3-*O*-glucoside-4-vinylcatechol	Mv-vcol	0.62	2.22	1.37	0.52	6.95	2.85	**0.037 ***
26	Malvidin-3-*O*-acetylglucoside-4-vinylphenol	MvA-vpol	0.55	4.75	1.96	0.09	5.42	1.86	0.719
27	Malvidin-3-*O*-acetylglucoside-4-vinylsyringol	MvA-vsol	0.11	1.53	0.59	0.1	1.53	0.59	0.28
28	Malvidin-3-*O*-acetylglucoside-4-vinylguaiacol	MvA-vgol	0.18	1.58	0.69	0.18	1.58	0.7	0.538
29	Malvidin-3-*O*-acetylglucoside-4-vinylcatechol	MvA-vcol	0.27	1.770	0.860	0.22	2.72	1.28	0.075
30	Malvidin-3-*O*-coumaroylglucoside-4-vinylphenol	MvC-vpol	0.22	1.94	1.13	0.34	2.51	1.25	0.191
31	Malvidin-3-*O*-coumaroylglucoside-4-vinylsyringol	MvC-vsol	0.1	1.53	0.58	0.1	1.53	0.58	**0.040 ***
32	Malvidin-3-*O*-coumaroylglucoside-4-vinylguaiacol	MvC-vgol	0.13	1.55	0.62	0.14	1.56	0.64	**0.004 ****
33	Malvidin-3-*O*-coumaroylglucoside-4-vinylcatechol	MvC-vcol	0.17	1.62	0.78	0.17	1.98	0.86	0.516
	**Total pinotins**	T-Pinontins	3.63	30.42	13.31	3.64	36.89	16.07	
34	Malvidin-3-*O*-glucoside-ethyl-catechin	Mv-e-cat	0.35	5.07	1.89	0.22	7.84	2.27	0.364
35	Malvidin-3-*O*-acetylglucoside-ethyl-(epi)catechin	MvA-e-(e)cat	0.16	2.03	1.03	0.11	2.01	1.04	0.893
36	Malvidin-3-*O*-glucoside-ethyl-di(epi)catechin	Mv-e-di(e)cat	0.14	1.6	0.72	0.11	1.64	0.8	0.167
37	Malvidin-3-*O*-coumaroylglucoside-ethyl-(epi)catechin	MvC-e-(e)cat	0.31	1.82	1.08	0.15	2.01	1.21	0.103
	**Total A-e-F**	T-A-e-F	0.95	10.52	4.72	0.59	13.51	5.31	
38	Malvidin-3-*O*-glucoside-4-vinyl(epi)catechin	Mv-v(e)cat	0.37	1.87	1.08	0.41	2.28	1.42	**0.009 ****
39	Malvidin-3-*O*-glucoside-4-vinyl-di(epi)catechin	Mv-vdi(e)cat	0.12	1.56	0.63	0.15	1.61	0.67	**0.014 ***
40	Malvidin-3-*O*-coumaroylglucoside-4-vinyl(epi)catechin	MvC-v(e)cat	0.16	1.6	0.68	0.19	1.72	0.74	**0.015 ***
41	Malvidin-3-*O*-acetylglucoside-4-vinyl(epi)catechin	MvA-v(e)cat	0.28	1.7	0.89	0.24	1.69	0.92	0.502
	**Total A-v-F**	T-A-v-F	0.94	6.73	3.29	0.98	7.3	3.76	
42	Malvidin-3-*O*-glucoside-di(epi)catechin	Mv-di(e)cat	0.11	1.53	0.6	0.1	1.54	0.6	**0.018 ***
43	(Epi)catechin-Malvidin-3-*O*-coumaroylglucoside	(E)cat-MvC	0.18	1.58	0.7	0.13	1.62	0.74	0.082
44	(Epi)catechin-Malvidin-3-*O*-acetylglucoside	(E)cat-MvA	0.43	1.7	0.95	0.18	1.7	0.91	0.198
45	(Epi)gallocatechin-Malvidin-3-*O*-glucoside	Mv-(e)gcat	0.10	1.53	0.59	0.09	1.53	0.58	0.121
46	Malvidin-3-*O*-glucoside-(epi)catechin (A type)	Mv-(e)cat	0.55	1.71	1.15	0.52	1.92	1.26	0.234
47	(Epi)catechin-Malvidin-3-*O*-glucoside	(E)cat-Mv	1.44	3.65	2.26	0.98	3.48	2.46	0.213
	**Total F-A/A-F**	T-F-A/A-F	2.81	11.7	6.25	2.00	11.79	6.55	
48	Catechin	C	11.02	100.26	45.6	14.95	64.72	40.34	0.323
49	Gallocatechin	GC	n.d.	0.43	0.13	n.d.	1.18	0.2	0.312
50	Epicatechin	EC	9.97	51.9	28.92	7.18	47.5	27.5	0.699
51	Epigallocatechin	EGC	n.d.	17.66	6.63	n.d.	16.24	5.05	0.067
52	Procyanidin B1	Pro B1	4.76	93.98	48.68	3.93	95.73	42.36	0.322
53	Procyanidin B2	Pro B2	3.82	71.09	24.58	1.56	70.29	27.42	0.363
	**Total flavan-3-ols**	T-Flavan-3-ols	29.58	335.32	154.54	27.61	295.67	142.86	
54	Protocatechuic acid	ProA	0.47	3.39	1.01	0.01	2.48	0.85	0.355
55	Gentisic acid	GenA	n.d.	0.54	0.06	n.d.	0.4	0.06	0.976
56	Vanillic acid	VanA	n.d.	4.02	1.09	n.d.	6.65	1.17	0.754
57	Gallic acid	GalA	10.82	38	21.68	n.d.	44.57	21.06	0.826
58	Chlorogentic acid	ChlA	n.d.	8.94	1.2	n.d.	6.69	1.42	0.631
	**Total hydroxybenzoic acids**	T-Hydroxybenzoic acids	11.29	54.87	25.03	0.01	60.79	24.56	
59	2-Hydroxycinnamic acid	2HydroA	n.d.	0.97	0.64	n.d.	0.98	0.6	0.375
60	3-Hydroxycinnamic acid	3HydroA	n.d.	1.22	0.72	n.d.	1.54	0.78	0.473
61	4-Hydroxycinnamic acid	4HydroA	0.33	0.95	0.79	0.31	1	0.81	0.767
62	Caffeic acid	CafA	n.d.	9.49	1.12	n.d.	28.09	8.72	**0.017 ***
	**Total hydroxycinnamates**	T-Hydroxycinnamates	0.33	12.63	3.27	0.31	31.61	10.9	
63	Myricetin	Myr	0.66	4.3	2.22	0.32	5.8	2.08	0.629
64	Myricetin-3-*O*-glucoside	Myr-3-glu	1.68	40.34	19.99	n.d.	53.97	19.96	0.991
65	Myricetin-3-*O*-galactoside	Myr-3-gal	0.06	1.49	0.45	n.d.	1.02	0.23	**0.007 ****
66	Kaempferol-3-*O*-glucoside	Kae-3-glu	0.24	2.36	0.85	0.21	1.22	0.6	0.219
67	Kaempferol-3-*O*-galactoside	Kae-3-gal	n.d.	0.57	0.25	n.d.	0.48	0.16	0.076
68	Quercetin-3-*O*-glucoside	Que-3-glu	n.d.	22.21	6.89	n.d.	12.08	4.33	0.233
69	Quercetin-3-*O*-glucuronide	Que-3-glucur	5.88	25.36	14.89	7.13	31.87	14.73	0.907
70	Quercetin-3-*O*-galactoside	Que-3-gal	0.13	1.65	0.89	0.07	1.22	0.5	**0.005 ****
71	Quercetin-rhamnoside	Que-rha	0.04	0.35	0.19	0.14	0.57	0.27	0.06
72	Dihydroquercetin	Dihydroque	n.d.	1.91	0.4	n.d.	3.74	0.61	0.536
	**Total flavonols**	T-Flavonols	8.69	100.54	47.03	7.86	111.98	43.47	

*^a^* n.d., not detected; *, **: Significant at *p* ≤ 0.05 and 0.01, respectively.

**Table 4 foods-12-01110-t004:** The paired *t*-test results of CIELAB chromatic parameters and sensory profiles in Cabernet Sauvignon and Marselan dry red wines from five regions in China.

Attributes	Cabernet Sauvignon			Marselan			*p*-Value
	Minimum	Maximum	Mean	Minimum	Maximum	Mean	
*L**	37.450	66.160	55.880	25.910	62.980	47.036	**<0.001 *****
*a**	29.790	53.960	39.134	36.580	61.760	45.918	**<0.001 *****
*b**	5.240	22.010	14.540	1.410	20.400	13.055	0.168
Red fruit	4.923	6.583	5.673	5.349	6.547	5.713	0.824
Black fruit	5.384	6.111	5.718	5.050	6.540	5.745	0.839
Fresh fruit	2.910	4.738	3.491	3.081	4.500	3.708	0.171
Floral	2.456	4.052	2.796	2.361	4.167	3.048	0.166
Dried fruit	2.057	3.771	2.847	1.589	4.343	3.020	0.099
Herbaceous	2.278	4.842	3.197	2.194	3.334	2.689	**0.013 ***
Sweet	1.383	2.623	1.756	1.403	3.118	2.089	0.076
Oak	2.291	5.041	3.233	1.972	4.174	2.939	**0.020 ***
Astringency	3.188	7.009	4.311	2.906	5.484	4.068	0.309
Tannin quality	4.268	6.673	5.725	5.238	6.574	5.936	0.275

*, ***: Significant at *p* ≤ 0.05 and 0.001, respectively.

## Data Availability

Data are contained within the article.

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
