# Peer review of "Comparison of Chemical and Sensory Profiles between Cabernet Sauvignon and Marselan Dry Red Wines in China"

_foods, 2023, doi:10.3390/foods12051110_

Round 1

Reviewer 1 Report

Statistical analysis section needs to be elaborated. What are the main aims of each statistical analysis here? Which one belongs to which dataset?

Have the authors assessed for normality here?

Table 1. I presume therefore the authors analysed 24 wines in total, but how come the analysis merged the C.S. and the Marselan variety and generalising it using t-tests? This doesn't really make sense as different regions and different wineries would have their own processing techniques which may rise differences between the samples?

I'm now seeing the individual samples here projected in PLSDA, so what is the Marselan and C.S. loading here? A supplementary variable? More details needed.

PLSDA was done for volatiles but not for phenolics, why?

Since it is a Descriptive Analysis, did the authors check for panel performance and ran some removals of panelists/attributes? 

It is still unclear to me how many samples being evaluated here?

Rather than quoting previous study [12], elaborate in a short manner on how it was carried out. 

Based on my understanding, the authors attempt to run t-test on sensory data - please replace this with ANOVA where it takes into account panel and replication effect and its interaction too. This is the golden standard in sensory eval.

Reviewer 2 Report

The manuscript by Song et al presents a comparison of chemical and sensory profiles between two grape verieties used in winemaking in China. Various analytical methods were used in the work, such as: HS-SPME-GC-MS, SPE-GC-QqQ-MS/MS, HPLC-QqQ-MS/MS, or colorimetric measurements which were properly described. 

The first remark concerns statistical methods. How to check if the assumptions necessary to use the paired t-test were met?

The constructed classification model (PLS-DA) should be further validated. Parameter values should be calculated: Sensitivity==TP/(TP+FN); Specificity=TN/(TN+FP); Accuracy = (TN + TP)/(TN + TP + FN + FP) where TP are true positives, TN are true negatives, FP are false positives, and FN are false negatives.

In Figures 1 and 2, I suggest consistently using the same markers and their colors and defining them on the legends.

Please consider including part of the results from Tables 2 and 3 in supplementary materials.

It is worth conducting a statistical analysis of the oenological parameters presented in Table S1. This will allow to assess whether they had any influence on the obtained results.

The authors meticulously discuss the obtained results by referring to the literature.

Even though, I don't feel qualified to judge about the English language and style, it seems that the English language needs correction. 

In my opinion, the manuscript needs minor revision.

Round 2

Reviewer 1 Report

I disagree with the authors approach on paired t-test, perhaps a classical ANOVA with multiple factors or MANOVA would work better here. 

OK it is clear that PLSDA were carried out but the variables do not make sense. The triangles are the X which corresponds to the metabolites, so how did the author projcet the blue squares and the red squares?

Running a t-test for sensory data again is inappropriate, do run an ANOVA using the classical sensory model and add a wine type factor here in your ANOVA model (with no interaction perhaps). Then, generalise the results from there.
